# Do aqueous solutions contain net charge?

Tao Ye[1,2]*, Gerald H. Pollack[1]

**1** Department of Bioengineering, University of Washington, Seattle, WA, United States of America,
**2** Department of Civil and Environmental Engineering, South Dakota School of Mines and Technology, Rapid City, SD, United States of America

* Tao.Ye@sdsmt.edu

## Abstract

Solutions with high pH values are sometimes thought to contain net negative charge because of an excess of $OH^-$ groups, while solutions with low pH values are thought opposite. To follow up on these speculations, we used a simple electrochemical cell to study three types of solution: electrolyzed waters with differing pH values; acids and bases with different pH values; and various salt solutions. When electrolyzed waters of various pH values were tested against water of pH 7, we found that acidic waters were indeed positively charged, while basic waters were negatively charged. We found much the same when standard acids and bases were compared to reference solutions: acidic solutions were positively charged while basic solutions were negatively charged. Various salts, including NaCl, KCl, $Na_2SO_4$, and $K_2SO_4$, were also tested against DI water (containing trace amounts of NaCl to lend conductivity). Surprisingly, all salts were found to be negatively charged, more so as their concentrations increased. This collection of results supports the hypothesis that at least some aqueous solutions may contain net charge.

## Introduction

Of all chemical measurements, perhaps the most common is pH. Yet, when pressed with the simple question, "does a high pH value imply net negative charge?" or "does a low pH value imply net positive charge?" naïve observers consistently respond in mixed fashion. Some argue that no solution can bear net charge; others opine that free $OH^-$ or $H^+$ indeed imply the presence of net charge. The question: which response is correct?

In the study reported here, we attempt to answer that question, and more.

If, indeed, the pH measurement implies net charge, then that net charge ought to be measurable. The obvious approach is to measure the electrical potential difference between otherwise identical solutions of high and low pH. If, indeed, the solutions are oppositely charged, then, an electrical potential difference should exist between the two of them.

Such measurements are facilitated by the advent of water electrolyzers. Designed as health expedients, electrolyzers split input water into waters with high and low pH values. High pH water is thought to produce health benefits [1–4], while the low pH water is commonly discarded. Here, both products are useful, not just because they provide disparate pH values for testing, but also because the substances produced are consistent: water, alone. Hence, an

**Competing interests:** The authors have declared that no competing interests exist.

opportunity exists to determine whether indeed high and low pH values of the same substance imply solutions with net negative and positive charge, respectively.

## Experimental section

Electrolyzed waters with different pH values (i.e., 2, 6, 7, 9, and 10) were produced by a water electrolyzer ($H_QO$ Next Generation Hydration, Q Sciences). To obtain solutions with different pH values, deionized water (DI water with a resistivity of 18.2 MΩ cm) was collected from a Barnstead D3750 Nanopure Diamond purification system. Solutions with different pH values (i.e., 2–14) were then prepared by adding concentrated hydrochloric acid (HCl) or sodium hydroxide (NaOH). Solutions with pH values near 7 were produced freshly by the DI water supply and used immediately.

Salt solutions, including sodium chloride (NaCl), potassium chloride (KCl), sodium sulfate ($Na_2SO_4$), and potassium sulfate ($K_2SO_4$) were prepared by dissolving the respective salts in DI water. $K_2SO_4$ ($\geq$ 99.0%) and potassium hydroxide (KOH, $\geq$ 85.0%, Supelco) were purchased from Sigma-Aldrich. Concentrated nitric acid ($HNO_3$, 68.0–70.0%) and HCl (36.5 to 38.0%) were supplied by Macron Fine Chemicals™. Other chemicals, including $Na_2SO_4$, NaCl, KCl, and NaOH, were obtained from Fisher Scientific.

A simple, home-made electrochemical cell was used to measure the potential difference between two solutions (Fig 1). One solution was added in the left beaker, the other in the right beaker. A salt bridge connected the two solutions. To fashion the bridge, natural white cotton-string twine (string diameter ~1.5 mm, manufacturer KingLake) was cut into 12-cm lengths,

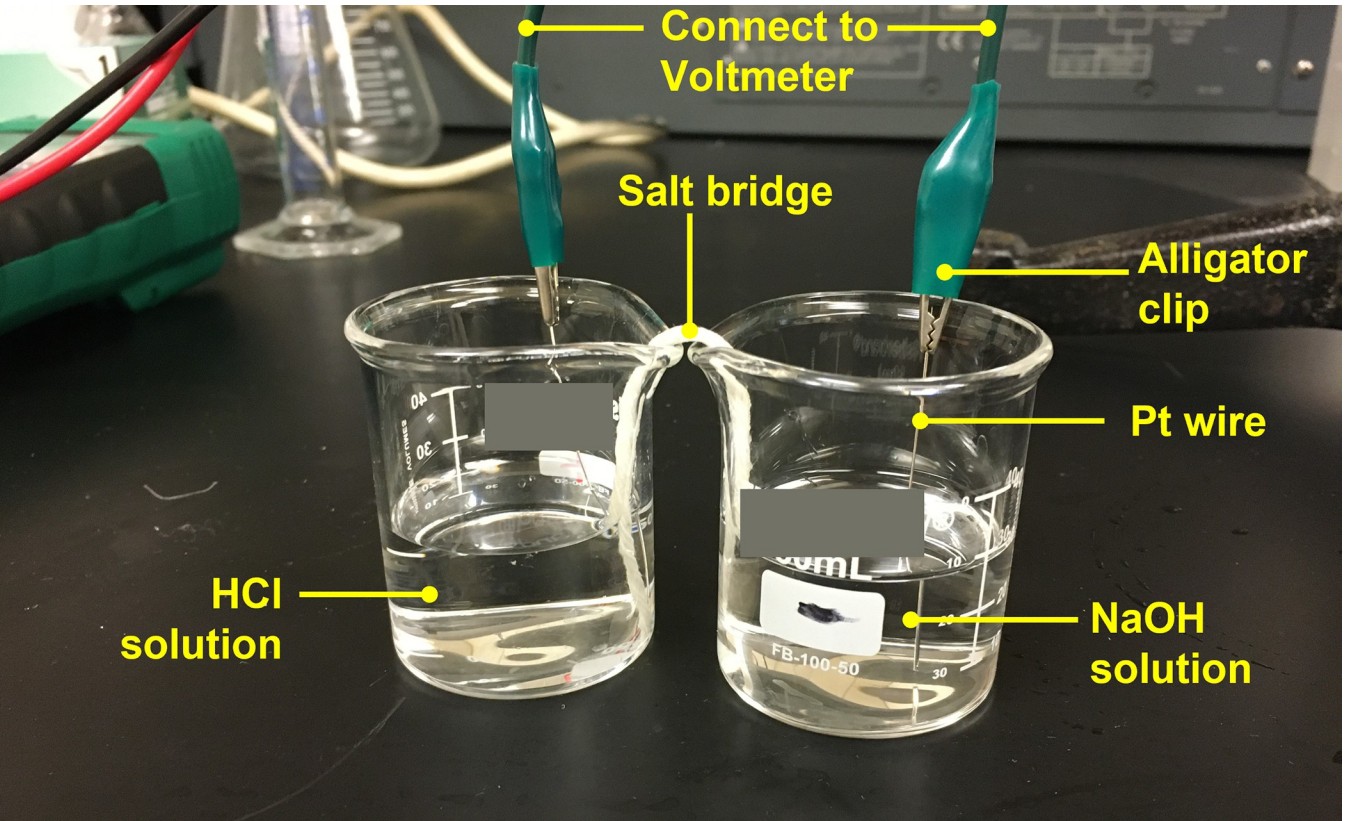

**Fig 1. A photo of the electrochemical cell.**

which were then soaked in saturated KCl solution. Once these strings were wetted and saturated thoroughly overnight, the eight strands (diameter ~0.19 mm) composing each string were separated, and each strand was used as a salt bridge. The insertion of a concentrated KCl aqueous solution between two dilute aqueous electrolyte solutions can effectively eliminate the liquid junction potential. Therefore, salt bridges were all soaked in saturated KCl before use. The potential difference was measured by a voltmeter (MS8268, Mastech). Platinum wires (diameter 0.5 mm, Alfa Aesar™, Premion™), or stainless steel wires (diameter 0.51 mm, Alfa Aesar™) were used as electrodes, connecting the respective solutions to the voltmeter.

In each experiment, 25 mL of solution was first added to each beaker. The electrodes were then immersed into the respective solutions. The voltmeter was then turned on, and the two solutions were immediately connected by the salt bridge. Voltages were recorded at 1 min, 3 min, and 6 min after connecting. All experiments were conducted at room temperature (~20 ± 1°C). Error bars in the figures are standard deviations, calculated from repeated experiments of at least three times.

## Results

### Control experiments on salt-bridge composition

For interconnecting the two solutions, we began experiments using 12-cm long cotton strings. However, potential differences up to 50–60 mV were observed between supposedly neutral solutions (i.e., 150 mM NaCl and 150 mM KCl), suggesting salt-bridge-related artifact. To eliminate this artifact, we divided these strings (diameter 1.5 mm) into constituent strands (diameter ~0.19 mm). The potential differences obtained by using these thinner strands were on the order of 1 mV, indicating nearly no salt-bridge-related artifact. Therefore, all experiments were conducted using these thinner strands.

### Electrolyzed water

We first conducted experiments with electrolyzed waters, i.e., waters emerging from the apparatus with selectable high and low pH values. Results are shown in **Fig 2**. Compared to water

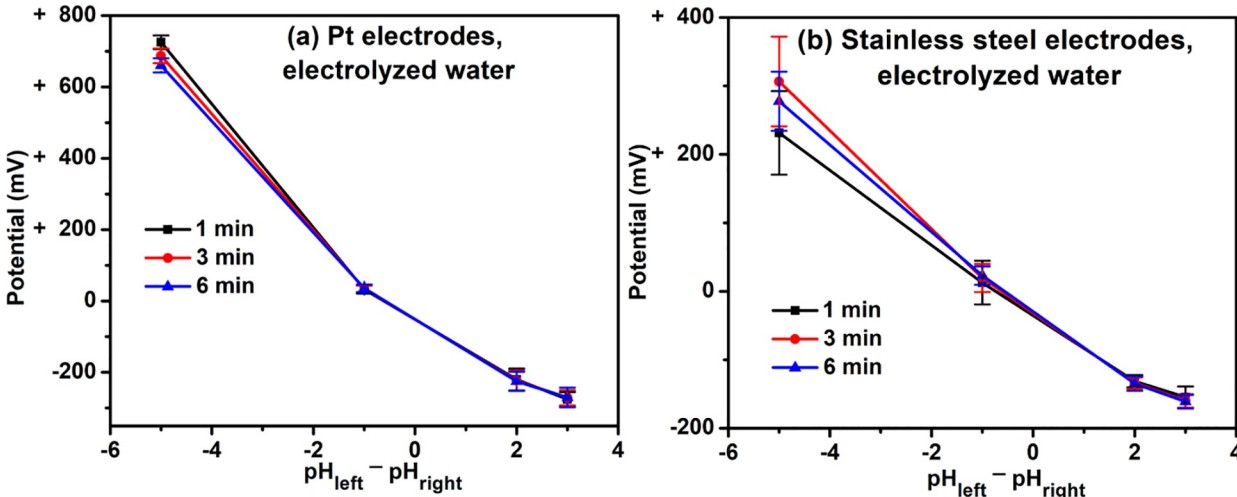

**Fig 2. Potential difference between solutions produced by a water electrolyzer.** Solutions in the right beaker (connected to the negative terminal of the voltmeter, **Fig 1**) were kept at pH 7, while solutions with pH 2, 6, 9, or 10 were placed in the left beaker. The $pH_{left}−pH_{right}$ indicates the pH value of the solution in the left beaker minus the pH value of the solution in the right beaker. For example, when the pH of the solution in the left beaker was 9, the $pH_{left}−pH_{right}$ value would be 2 (i.e., 9 minus 7).

of pH 7 (produced by the electrolyzer), acidic waters were positively charged, while basic waters were negatively charged. For example, acidic water (pH 2; $pH_{left}-pH_{right}$ = -5) showed a potential difference as high as +725 mV, while that of basic water (pH 10; $pH_{left}-pH_{right}$ = 3) was -275 mV (Fig 2A). The potential differences measured using platinum electrodes showed similar trends as with stainless steel electrodes (Fig 2A & 2B).

## Acids and bases

We also tested the electrical potential differences between common acids and bases (Fig 3 & Tables 1 and 2). Acids (pH 2 and 4, HCl), DI water, and bases (pH 10, 12, and 14, NaOH) were tested vs. reference solutions (pH 4 [HCl] or pH 10 [NaOH]). For experiments with reference solution of pH 4, when the pH of the test solution was higher than 4, the measured potential difference was negative; when the pH was 4, the potential became nearly zero; and, when the pH was lower than 4, the potential was positive (Fig 3A & 3B). Similar trends were observed for reference solution of pH 10. And, results were essentially independent of the type of electrode used.

Overall, these results indicate that a solution that is more acidic than the reference solution is more positively charged than the reference solution, while a solution that is more basic than the reference solution is more negatively charged.

Further, common acids (HCl and $HNO_3$) and bases (NaOH and KOH) were tested individually relative to neutral pH solutions, i.e., to solutions near pH 7. Ideally, DI water should be used for the latter; however trace amounts NaCl were added in order to impart the required conductivity. Among the three concentrations tested (i.e., 1 mM, 1 μM, and 1 nM), the results obtained using 1 μM NaCl were the most consistent, with small standard deviations (data not shown). 1 μM NaCl solution was then used for further experiments testing common acids and bases against neutral pH.

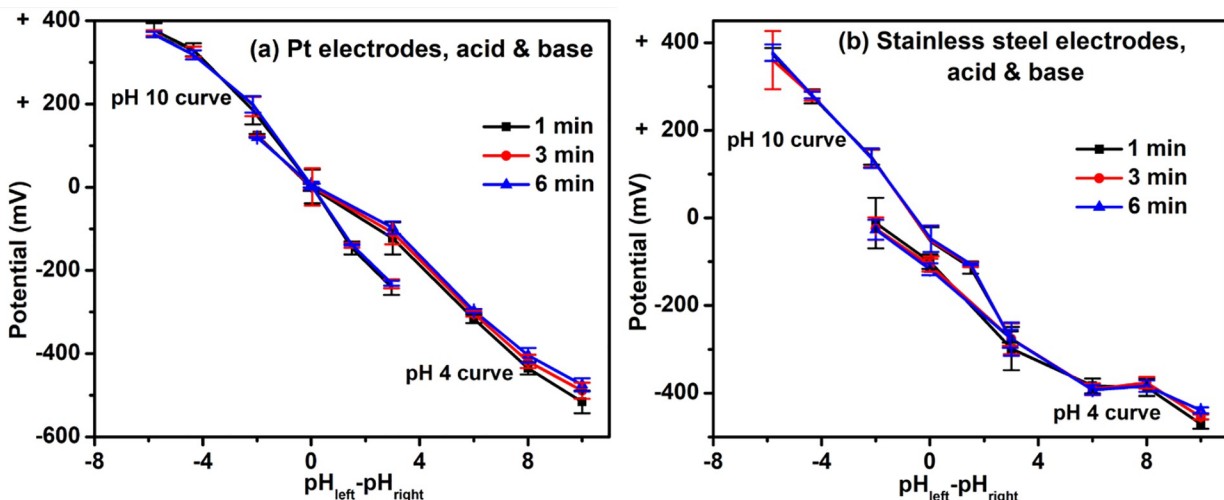

**Fig 3. Potential difference between acids and bases.** Solutions of various pH values were prepared by adding varied amounts of DI water to concentrated HCl or 1 M NaOH. For the experiments at reference value of pH 4, solutions of pH 4 in one beaker (right in Fig 1) were connected to the negative terminal of the voltmeter, while solutions with various pH values (i.e., 2, 4, 7, 10, 12, and 14) were added to the left beaker, which was in turn connected to the positive terminal of the voltmeter. Similarly, for experiments at pH 10, the solutions in the right beaker were maintained at pH 10. The $pH_{left}-pH_{right}$ indicates the pH value of the solution in the left beaker minus the pH value of the solution in the right beaker. For example, when the pH of the solution in the left beaker was 12, and the pH of the solution in the right beaker was 10, the $pH_{left}-pH_{right}$ was 12 minus 10, or 2.

**Table 1. (a) Electrical potential differences between acids (HCl) with differing pH values and neutral solutions (containing 1μM NaCl); and, bases (NaOH) and neutral solutions (containing 1μM NaCl).** Measurements taken at different times to test stability[a]. **(b)** Electrical potential differences between acids (HNO₃) and neutral solutions (containing 1 μM NaCl); and, bases (KOH) and neutral solutions (containing 1 μM NaCl)[a].

| pH | Potential (mV) | | |
|---|---|---|---|
| | 1 min | 3 min | 6 min |
| 2 | +276.9 ± 11.0 | +292.5 ± 6.9 | +294.1 ± 14.4 |
| 4 | +172.4 ± 7.3 | +178.5 ± 6.2 | +185.2 ± 5.7 |
| 10 | -205.9 ± 20.0 | -212.3 ± 7.6 | -211.4 ± 6.0 |
| 12 | -269.1 ± 40.0 | -259.5 ± 33.4 | -250.2 ± 24.2 |
| 14 | -386.2 ± 5.9 | -386.2 ± 6.9 | -380.3 ± 6.1 |

| pH | Potential (mV) | | |
|---|---|---|---|
| | 1 min | 3 min | 6 min |
| 2 | +202.6 ± 3.5 | +194.9 ± 4.4 | +186.8 ± 7.2 |
| 4 | +81.3 ± 17.1 | +71.8 ± 10.9 | +63.1 ± 7.7 |
| 10 | -219.8 ± 16.9 | -222.2 ± 11.3 | -223.2 ± 11.8 |
| 12 | -302.2 ± 19.0 | -300.4 ± 20.4 | -294.3 ± 21.4 |
| 14 | -380.5 ± 11.7 | -371.5 ± 8.9 | -364.5 ± 14.7 |

[a] Acids (pH 2 and 4) and bases (pH 10, 12, and 14) were prepared by adjusting the pH values of concentrated HCl or 1 M NaOH by diluting with DI water. The acid solutions were added in the beaker that was connected to the positive terminal of the voltmeter, while the neutral solution (containing 1μM NaCl) was connected to the negative terminal. Stainless steel wires were used as electrodes.

[a] Experiments were conducted similarly to those tabulated in **Table 1A**, except that concentrated HNO₃ and KOH were used to prepared acids and bases, respectively.

Again, acids and bases were respectively confirmed to be charged positively and negatively (**Tables 1 and 2**). Generally, the more acidic the solution, the more positive was the charge. For example, HCl solution at pH 2 showed a potential difference of +294 mV, while that same acid at pH 4 was +185 mV (**Table 1A**). Similarly, the more basic the solution, the more negative was the charge. For example, NaOH solution at pH 10 showed a potential difference of -211 mV, while NaOH at pH 14 gave -380 mV (**Table 1A**). To guard against any electrode-dependent effects, both platinum electrodes and stainless steel electrodes were used, and similar results were obtained (**Tables 1 and 2**). We noticed that in the pH 10 tests (Tables 1 & 2), the potentials registered fluctuated and the difference from 1–6 min was not as significant as other pHs that were tested. We are not sure what is the exact reason behind it. It can be that the ionic strength or pH difference between the left beaker solution (positive terminal, pH 10, NaOH) and the right beaker solution (negative terminal, 1 μM NaCl) was not significant.

## Salts

Different salts including NaCl, KCl, Na₂SO₄, and K₂SO₄ were also tested against DI water (containing trace amounts of NaCl) to check for potential differences. All salts were found to be negatively charged, irrespective of concentration (**Figs 4 & 5**). Typically, salts became more negatively charged as their concentrations increased. At high salt concentrations (> 1 mM), NaCl seemed to be more negatively charged than KCl (**Fig 4**). For example, 3 M NaCl showed an electrical potential of -181 mV, while 3 M KCl was only -133 mV. Solutions of Na₂SO₄ showed similar potentials to those of K₂SO₄. Note that the modest potentials measured at low concentrations should not be considered reliable because the test-salt concentrations were equal to or even lower than those of the "neutral pH" beaker.

**Table 2. Platinum electrodes replacing stainless steel electrodes. (a)** Electrical potentials between acids (HCl) and neutral solutions (containing 1μM NaCl), and bases (NaOH) and neutral solutions (containing 1μM NaCl)[a]. **(b)** Electrical potentials between acids (HNO₃) and neutral solutions (containing 1μM NaCl), and bases (KOH) and neutral solutions (containing 1μM NaCl)[a].

| pH | Potential (mV) | | |
|---|---|---|---|
| | **1 min** | **3 min** | **6 min** |
| 2 | +192.3 ± 7.1 | +182.3 ± 2.7 | +177.7 ± 0.8 |
| 4 | +85.7 ± 3.8 | +79.3 ± 1.2 | +75.4 ± 1.9 |
| 10 | -199.8 ± 2.7 | -199 ± 3.6 | -195.7 ± 4.5 |
| 12 | -317 ± 12.1 | -321.7 ± 5.7 | -322.8 ± 1.8 |
| 14 | -382.6 ± 18.9 | -374.2 ± 16.0 | -378.4 ± 12.5 |

| pH | Potential (mV) | | |
|---|---|---|---|
| | **1 min** | **3 min** | **6 min** |
| 2 | +158.8 ± 6.3 | +155.3 ± 6.4 | +153.4 ± 3.9 |
| 4 | +72.5 ± 4.8 | +67.2 ± 2.7 | +63.3 ± 2.1 |
| 10 | -213.5 ± 6.7 | -211.2 ± 2.9 | -207.1 ± 2.3 |
| 12 | -318.8 ± 11.1 | -316.4 ± 8.7 | -312.7 ± 9.7 |
| 14 | -374.8 ± 8.3 | -374.8 ± 8.8 | -382.5 ± 13.9 |

[a] Experiments were conducted similarly to those in Table 1A, except that Pt wires were used as electrodes.

[a] Experiments were conducted similarly to those in Table 2(A), except that concentrated HNO₃ and KOH were used to prepare acids and bases, respectively.

## Discussion

The results confirm that aqueous solutions may well bear net charge. Using waters obtained from a standard electrolyzer, we found a substantial difference of electrical potential between pairs of waters that were identical except for differences of pH value. To a first approximation, the potential difference was directly proportional to the pH difference between the two solutions. Hence, pH values appear to be measures of electrical potential, which in turn imply that the respective solutions contain net charge.

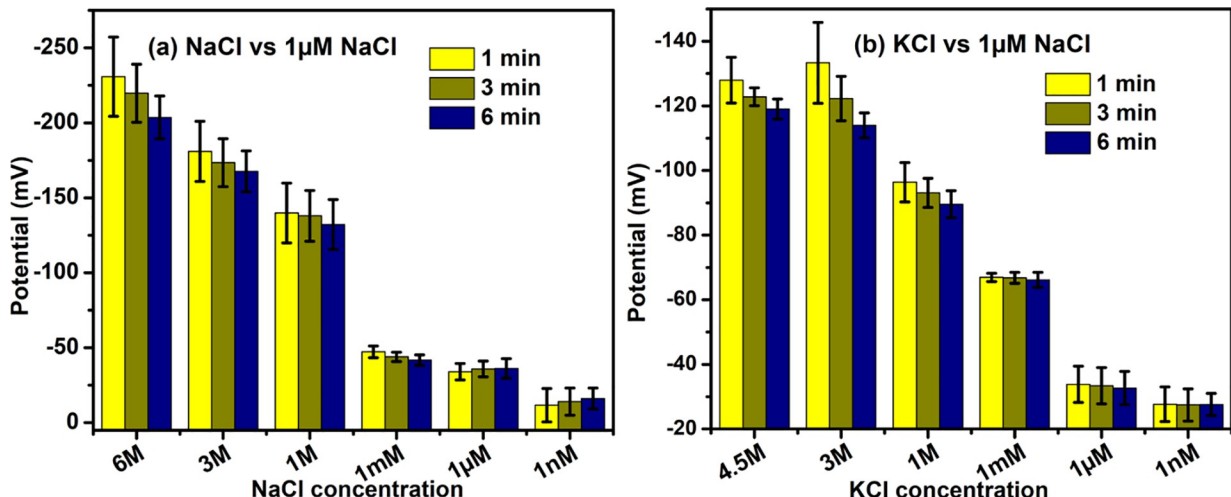

**Fig 4. Potential differences between common salt solutions and DI water (containing 1 μM NaCl).** (a) NaCl solutions of different concentration in one beaker (left in Fig 1) were connected to the positive terminal of the voltmeter, while neutral solutions were contained in the other beaker (right in Fig 1), which was connected to the voltmeter's negative terminal. (b) Experiments with KCl were arranged similarly. Stainless steel wires were used as electrodes.

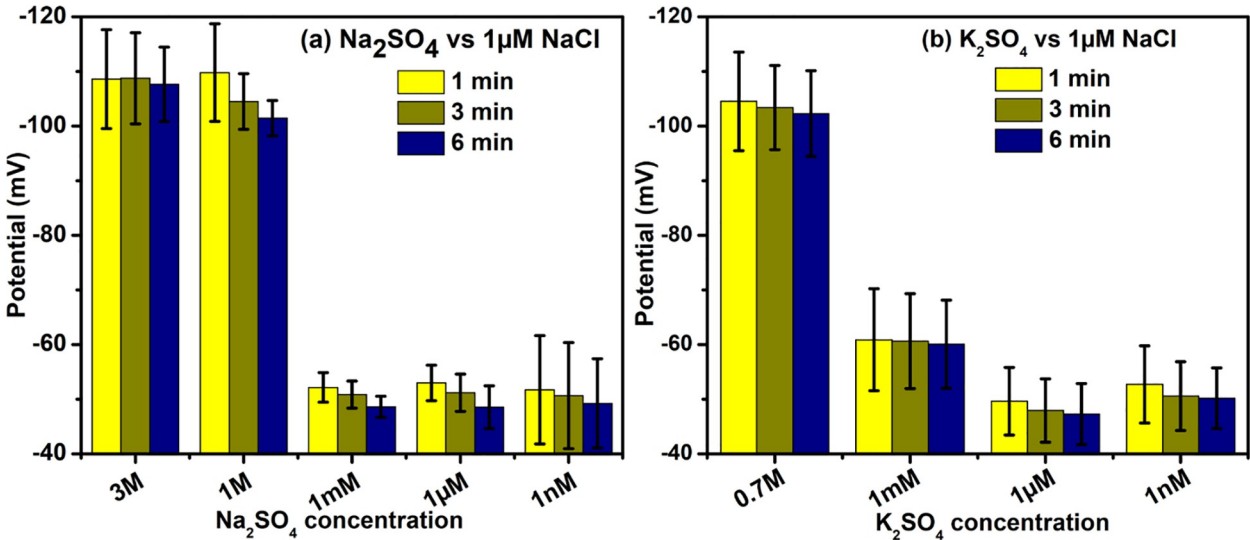

**Fig 5. Similar to Fig 4, but with different salts.** Stainless steel wires were used as the electrodes.

## Controls

Two important control experiments were critical to the interpretation.

First, the electrodes. The experimental results could possibly arise artificially from reactions at electrode surfaces with adjacent water molecules [5,6]. Such potential artifacts are well recognized [7,8]. To avert any such artifacts, we employed electrodes that are expected to be among the least reactive: platinum. We then repeated measurements using stainless steel electrodes of quality commonly used in medical applications. Results were similar (e.g., **Fig 2A vs. 2B**). Hence, the results do not appear to arise from some kind of metal-water interaction or any specifics associated with a particular electrode. They appear to be more general.

A second potential artifact could arise from the salt bridge interconnecting the two beakers of water. Such bridges are necessary to complete the electrical circuit. Without a conductor between the two solutions, no current can flow; hence no meaningful potential difference can be registered. In this bridging capacity, metal conductors commonly fail because of reactions between metals and salts contained within the experimental chambers; hence, replacement with so-called salt bridges have become standard. Standard bridges comprise cotton strings, soaked in saturated KCl solution [9].

Preliminary experiments with salt bridges revealed an odd result: Even with a pH difference of zero between the two beakers, we found an appreciable net potential difference. The difference could logically be ascribed to the bridge itself, which consisted of a multi-stranded cotton string saturated with KCl solution. Each string contained eight smaller intertwined strands (diameter 0.19 mm). In the hope that the potential difference might vanish over time, we tracked the time course over periods of six minutes, observing only modest change. Finally, we could solve the problem by altering the composition of the bridge, from the stout cotton string soaked in KCl to only a single 0.19 mm strand soaked in KCl. The procedure paid dividends: the potential differences diminished to negligibly small values, on the order of 1 mV. Thus, all reported experiments were conducted using this single-stranded bridge, ensuring that results were not likely to be contaminated by electrical potentials associated with the salt bridge.

## Electrolyzed water

With proper controls in place, we pursued experiments with electrolyzed water, the results of which support the notion that the pH value of a solution is a measure of its net charge. **Fig 2**

shows a plot of electrical potential difference vs. pH difference. Whether measured using platinum or stainless steel electrodes, the potential difference varied linearly with the pH difference. Both curves passed close to the zero—zero point, indicating no potential difference when the solutions had the same pH values. Since electrical potential is a measure of charge, we take this finding as evidence that pH is a measure of solution charge. In other words, the measured pH value is a reflection of net charge of a solution, or of water itself.

This conclusion should be unsurprising, given the nature of the pH measurement. Essentially, pH meters measure the electrical potential difference between the investigated solution and a reference solution [10]. Through an established scale, that voltage is converted to a corresponding pH value. In other words, the primary measurement is not pH; it's potential difference. Hence, the meter measurement itself is suggestive that charges accumulate in the respective flasks.

An obvious question is how charges could accumulate in a solution without getting quickly neutralized by oppositely charged ions in the air. While the answer is not completely clear, situations could exist whereupon the end result is not necessarily neutralization. One possibility is the air-water interface, whose structure could limit exchange between air and solution [11]. That could help sustain any solution charge.

A second possibility rests on water's dipolar structure. Consider two simple dipoles. Apart from the forces of thermal motion, those dipoles will tend to cling side-by-side with opposite charges adjacent to one another. Now, imagine if each "dipole" were to contain, say, one extra positive charge. With two positives at one end, and only one negative at the other, those "dipoles" should still cling, and many such clinging dipoles, side by side and end to end, could create a liquid-crystalline aggregate with net positive charge. If the clinging tendency is rather weak, then the aggregate could remain liquid-like, although likely of elevated viscosity because of the inter-molecular attractions.

Aggregates such as those could contain ample, sustained net charge. And, even if not aggregated in that manner, solution charges could still be prevented from mixing with the air by features of the air-water interface. Hence, neutralization is not a *necessary* condition for solutions. At least in theory, conditions may exist in which neutralization is not an obligatory outcome.

## Acids and bases

Besides the measurements made with electrolyzed water, we measured the electrical potential differences between common acids and bases. Again, the results showed potential differences, acids more positive than the bases (**Fig 3** & **Tables 1 and 2**). Ordinarily, such potential differences might be considered less reliable than those made using electrolyzed water because the substances in the respective flasks differ; one could argue for some differential impact of those substances on the electrodes, leading to an artifactual potential differences.

However, two arguments imply no such artifact. First, the results were similar when measured using different sets of metal electrodes, platinum and stainless steel. That spurious electrode interactions might yield the same artifact in distinct metals seems unlikely. Second, the results fit nicely with results obtained using the electrolyzed water, where the "acids" and "bases" had the same chemical composition: water. Hence, we think it unlikely that the acid-base results are based on artifact. Rather, consistent with the results obtained with the electrolyzer, acids appear to bear positive charge, bases negative charge.

The results may provide some understanding why many acids and bases are as reactive as they are, and why the combination can even react explosively. Opposite-charge polarities may facilitate interaction, i.e., the reactants may be drawn actively toward one another, causing powerful interactions.

The results also help explain the behaviors of pH-sensitive dyes, such as those in ordinary litmus paper. The standard explanation for color change lies in the dyes' supposed sensitivity to hydrogen-ion concentration. While such an explanation can make sense, the question arises: how does a dye molecule know whether a hydrogen ion does or does not lie nearby? Simpler is an interpretation based on environment: Charges produce electric fields, which may be felt at appreciable distances from the charge itself. Hence, the dye may change color based on the general environment, rather than on the presence of a particular ($H^+$) molecule nearby.

## Salts

Finding that acids and bases contain net charge, we wondered whether salts might contain net charges as well. We might not envision such, because salts are considered neutral. For example, NaCl should contain one positive (Na) and one negative (Cl) charge, yielding a neutral molecule, at least in theory. Nevertheless, we proceeded to test various salt solutions.

The results were surprising. All salts tested, NaCl, KCl, $Na_2SO_4$ and $K_2SO_4$, showed negative electrical potentials, especially at high concentration. We appreciate that, according to the periodic table, these salt solutions should contain no charge. However, the measurements showed otherwise. One possible interpretation is that negatively charged exclusion zone (EZ) water may form around salt molecules [11]. If the complementary positive charges were to escape through evaporation, then the solution would be left with net negative charge, as measured. It remains for future experiments to sort out these options.

## Conclusion

Prompted by the inability of students of science to respond uniformly to the question whether acids do or do not contain net positive charge, we sought a definitive answer. It appears that acids do contain net positive charge, while bases contain net negative charge.

How this happens remains incompletely clear. If true, however, then this finding would seem to have broad implication for chemical and biological science, for charges inevitably produce forces, and such forces would need to be taken into account in attempting to understand the nature of chemical and biological reactions.

The results obtained with salt solutions add to the enigma. Salt solutions are presumed neutral. However, the measurements reported here imply otherwise. If salt solutions genuinely bear charge, then the impact of these charges will likewise need to be considered in interpreting the results of many chemical reactions.

## Supporting information

**S1 Graphical abstract.**
(TIF)

## Acknowledgments

We thank Stan Esecson (YNR Marketing) for the generous loan of the electrolyzer. This work was supported by the SAGST Foundation, and by an anonymous donor.

## Author Contributions

**Conceptualization:** Gerald H. Pollack.

**Data curation:** Tao Ye.

**Formal analysis:** Tao Ye.

**Funding acquisition:** Gerald H. Pollack.

**Investigation:** Tao Ye.

**Methodology:** Gerald H. Pollack.

**Supervision:** Gerald H. Pollack.

**Writing – original draft:** Tao Ye, Gerald H. Pollack.

**Writing – review & editing:** Tao Ye, Gerald H. Pollack.

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
