## [Decision Letter · Decision Letter 0]

23 Mar 2022

PONE-D-22-00156Do aqueous solutions contain net charge?PLOS ONE

Dear Dr. Ye,

Thank you for submitting your manuscript to PLOS ONE. After careful consideration, we feel that it has merit but does not fully meet PLOS ONE’s publication criteria as it currently stands. Therefore, we invite you to submit a revised version of the manuscript that addresses the points raised during the review process.

ACADEMIC EDITOR: The major revision is required on the manuscript. 

Please submit your revised manuscript soon May 07 2022 11:59PM. If you will need more time than this to complete your revisions, please reply to this message or contact the journal office at plosone@plos.org. Please include the following items when submitting your revised manuscript:A rebuttal letter that responds to each point raised by the academic editor and reviewer(s). You should upload this letter as a separate file labeled 'Response to Reviewers'.A marked-up copy of your manuscript that highlights changes made to the original version. You should upload this as a separate file labeled 'Revised Manuscript with Track Changes'.An unmarked version of your revised paper without tracked changes. You should upload this as a separate file labeled 'Manuscript'.

We look forward to receiving your revised manuscript.

Kind regards,

Ajaya Bhattarai

Academic Editor

PLOS ONE

Journal Requirements:

(Please include your amended statements within your cover letter; we will change the online submission form on your behalf."

 NO - Include this sentence at the end of your statement: The funders had no role in study design, data collection and analysis, decision to publish, or preparation of the manuscript.)

Additional Editor Comments:

One reviewer suggested for major revision and another is rejected. The academic editor wants from the author to revise the manuscript.

Reviewers' comments:

Reviewer's Responses to Questions

**Comments to the Author**

1. Is the manuscript technically sound, and do the data support the conclusions?

Reviewer #1: Partly

Reviewer #2: Partly

2. Has the statistical analysis been performed appropriately and rigorously? 

Reviewer #1: I Don't Know

Reviewer #2: N/A

3. Have the authors made all data underlying the findings in their manuscript fully available?

Reviewer #1: No

Reviewer #2: Yes

4. Is the manuscript presented in an intelligible fashion and written in standard English?

Reviewer #1: No

Reviewer #2: Yes

5. Review Comments to the Author

Reviewer #1: 1. The title of the manuscript gives an impression that we are asking the question to the Google engine. Isn’t it possible to change the title with more appealing pattern?

2. In Abstract at “ ..solutions that were more acidic were more positively charged, while solutions that were more basic were more negatively charged”, my suggestion to change it with “ acidic solutions were positively charged while basic solutions were negatively charged”.

3. Normally we use the salt solution having the cations and anions which have similar mobility in solution but different from the ions present in our test solutions. But in this manuscript, sodium chloride (NaCl), potassium chloride (KCl), sodium sulfate (Na2SO4), and potassium sulfate (K2SO4) were used while as the salt bridge cotton string soaked with saturated KCl solution was used. If my belief is wrong, I want to see the logic behind it.

4. Regarding the data presented in Tables 1, 2: Just check the Table 1 (a) of electrical potential differences between acids (HCl) with differing pH values at different times to test stability. In cases of pH values 2 and 4, the potentials were found increased with increasing time from 1 min. to 6 min. while in cases of pH values 12 and 14 the potentials were found in decreasing order with time elapsed (though not sharply in case of pH 14). Such type of order was not obeyed in case of the solution having pH 10.

In another case the data shown in Table 1 (b) regarding the electrical potential differences between acids (HNO3) and neutral solutions at different times to test stability. The potentials were found decreased with increasing time from 1 min. to 6 min. in cases of the solutions with pH values 2, 4, 12 and 14 but why not the trend was followed in case of the solution having pH 10?

5. The results regarding the salt solutions of NaCl, KCl, Na2SO4 and K2SO4. Authors have mentioned that all these salt solutions at high concentrations had negative electrical potentials. The first logic gave for such controversial result as “the periodic table may be incomplete, and that some elements may themselves bear charge”. This logic is much more controversial than the result mentioned. My suggestion to Authors that prior to write such paper why don’t they redefine the Periodic Table and publish their logic?

6. I suggest the authors to see the book: Bickmore B.R., Wander M.C.F. (2018) Aqueous Solutions. In: White W.M. (eds) Encyclopedia of Geochemistry. Encyclopedia of Earth Sciences Series. Springer, Cham. https://doi.org/10.1007/978-3-319-39312-4_3

Reviewer #2: After going through the manuscript I found it to interesting but the authors reported only some data of their experimental work and based on these data they claimed the title of the manuscript to taken as granted scientifically, which according to me will not be scientifically justified right now. The work is premature and needs more extensive experimental and theoretical basis before the titled remark becomes scientifically established.

6. PLOS authors have the option to publish the peer review history of their article (what does this mean?). If published, this will include your full peer review and any attached files.

Reviewer #1: No

Reviewer #2: No

---

## [Author Response · Author response to Decision Letter 0]

20 Apr 2022

Response to Reviewers:

Do aqueous solutions contain net charge?

Reviewer 1

1. The title of the manuscript gives an impression that we are asking the question to the Google engine. Isn’t it possible to change the title with more appealing pattern? We thank the reviewer for this suggestion. We agree with the reviewer that this is a general question that people may search answers using Google. This is the reason why we believe this question is very important. We specifically discussed this situation in the first paragraph in Introduction. Therefore, we believe this simple and straightforward title may be a good choice. 

2. In Abstract at “..solutions that were more acidic were more positively charged, while solutions that were more basic were more negatively charged”, my suggestion to change it with “acidic solutions were positively charged while basic solutions were negatively charged”. We agree with the reviewer’s suggestion. The sentence was changed in the Abstract.

“We found much the same when standard acids and bases were compared to reference solutions: acidic solutions were positively charged while basic solutions were negatively charged.”

3. Normally we use the salt solution having the cations and anions which have similar mobility in solution but different from the ions present in our test solutions. But in this manuscript, sodium chloride (NaCl), potassium chloride (KCl), sodium sulfate

(Na2SO4), and potassium sulfate (K2SO4) were used while as the salt bridge cotton string soaked with saturated KCl solution

was used. If my belief is wrong, I want to see the logic behind it. Thank you for the reminder.

We partly agree with the reviewer. We tried to Google what salt bridge solution we should use before we conducted the experiments. We found KCl very common (https://en.wikipedia.org/wiki/Salt_bridge). 

We probably all know that the insertion of a concentrated KCl aqueous solution between two dilute aqueous electrolyte solutions can effectively eliminate the liquid junction potential (LJP), but there has virtually been no choice but a concentrated aqueous potassium chloride solution to minimize the LJP between two aqueous electrolyte solutions of different compositions.1

“Normally we use the salt solution having the cations and anions that have similar mobility in solution”. We are testing four: NaCl, KCl, Na2SO4, and K2SO4 solutions, hence, KCl should be an ideal salt solution because it has similar mobility to these solutions.

On-going research in our lab will consider using other salts as suggested by the reviewers. 

The following sentence is added in the manuscript.

“The insertion of a concentrated KCl aqueous solution between two dilute aqueous electrolyte solutions can effectively eliminate the liquid junction potential. Therefore, salt bridges were all soaked in saturated KCl before use.”

4. Regarding the data presented in Tables 1, 2: Just check the Table 1 (a) of electrical potential differences between acids (HCl) with differing pH values at different times to test stability. In cases of pH values 2 and 4, the potentials were found increased with increasing time from 1 min. to 6 min. while in cases of pH values 12 and 14 the potentials were found in

decreasing order with time elapsed (though not sharply in case of pH 14). Such type of order was not obeyed in case of the solution having pH 10.

In another case the data shown in Table 1 (b) regarding the electrical potential differences between acids (HNO3) and neutral solutions at different times to test stability. The potentials were found decreased with increasing time from 1 min. to 6 min. in cases of the solutions with pH values 2, 4, 12 and 14 but why not the trend was followed in case of the solution having pH 10? We thank the reviewer for the comment. 

We noticed it as well that in the pH 10 tests (Tables 1 & 2), the potentials registered fluctuated and the difference from 1-6 min was not as significant as other pHs that were tested. 

The phenomenon is very interesting. We are not sure what is the exact reason behind it. It can be that the ionic strength or pH difference between the left beaker solution (positive terminal, pH 10, NaOH) and the right beaker solution (negative terminal, 1 µM NaCl) was not significant. Or as suggested in previous comment, it can be the salt bridge solution. 

But the authors would like to mention that the potential registered may eventually stabilize, although it may take a long time.

Further research will be done in our on-going project.

The following sentences were added in the manuscript.

“We noticed that in the pH 10 tests (Tables 1 & 2), the potentials registered fluctuated and the difference from 1-6 min was not as significant as other pHs that were tested. We are not sure what is the exact reason behind it. It can be that the ionic strength or pH difference between the left beaker solution (positive terminal, pH 10, NaOH) and the right beaker solution (negative terminal, 1 µM NaCl) was not significant.”

5. The results regarding the salt solutions of NaCl, KCl, Na2SO4 and K2SO4. Authors have mentioned that all these salt solutions at high concentrations had negative electrical potentials. The first logic gave for such controversial result as “the

periodic table may be incomplete, and that some elements may themselves bear charge”. This logic is much more controversial than the result mentioned. My suggestion to Authors that prior to write such paper why don’t they redefine the Periodic Table and publish their logic? Plans are actually underway for one of the authors (GP) to write a book on the subject. Although the book draft is nearly complete, we expect, from experience, two to three years before it is finally published. Evidence will be presented to show that not all entries in the periodic table are necessarily correct, and that some elements may actually bear charge. Here, we merely allude to that possibility. The current work seems to us to be important enough that we hesitate to wait several years before it can be published.

6. I suggest the authors to see the book: Bickmore B.R., Wander M.C.F. (2018) Aqueous Solutions. In: White W.M. (eds)

Encyclopedia of Geochemistry. Encyclopedia of Earth Sciences Series. Springer, Cham. https://doi.org/10.1007/978-3-319-

39312-4_3 We thank the reviewer for the suggestion. 

We also thank the reviewer for the time and effort on reviewing the manuscript.

Reviewer 2

After going through the manuscript I found it to interesting but the authors reported only some data of their

experimental work and based on these data they claimed the title of the manuscript to taken as granted scientifically, which according to me will not be scientifically justified right now. The work is premature and needs more extensive experimental

and theoretical basis before the titled remark becomes scientifically established. We thank the reviewer for the time and effort on reviewing the manuscript. Yes, from experience, we do understand that gaining community acceptance will take time.

Nevertheless, we believe we have included all the data necessary to draw the conclusions that we draw. 

As mentioned in the first paragraph of the Introduction, “does a high pH value imply net negative charge?” or “does a low pH value

imply net positive charge?” are questions that consistently puzzle people. We believe our manuscript answers these questions scientifically, although we agree that it will take time for the scientific community to evaluate, and hopefully accept our conclusions.

1. Takashi, K.; Takahiro, Y., A New Salt Bridge Based on the Hydrophobic Room-Temperature Molten Salt. Bulletin of the Chemical Society of Japan 2006, 79, (7), 1017-1024.

---

## [Editor Report · Decision Letter 1]

9 May 2022

PONE-D-22-00156R1

Do aqueous solutions contain net charge?

PLOS ONE

Dear Dr.Tao Ye,

Thank you for submitting your manuscript to PLOS ONE. After careful consideration, we have decided that your manuscript does not meet our criteria for publication and must therefore be rejected.

Specifically, I am sorry that we cannot be more positive on this occasion, but hope that you appreciate the reasons for this decision.

Kind regards,

Ajaya Bhattarai

Academic Editor

PLOS ONE

Additional Editor Comments:

Comments from the reviewer

Dear Professor Bhattarai,

Regarding the manuscript which I reviewed earlier and then you sent to me as the revised version, I still have many objections some of which are:

1. Regarding the quarry of title: Though not satisfied with the logic of Author, it doesn't make any sense to deny for publication on such basis.

2. Correction done in accordance with my suggestion.

3. Regarding the use of salt solution in salt bridge: I am not satisfied at all. Furthermore writing "On-going research in our lab will consider using other salts as suggested by the reviewers." can't be taken as the positive token to consider the present manuscript.

4. Regarding electrical potential difference: I am not satisfied with the explanation as "Further research will be done in our on-going project.

5. Regarding controversial statement related with Periodic Table: The author has mentioned as "Plans are actually underway for one of the authors (GP) to write a book on the subject. Although the book draft is nearly complete, we expect, from experience, two to three years before it is finally published." On this ground if we accept author's logic right now, it can be disaster.

Thus my final decision on this manuscript is, Please REJECT it and with such controversial logic it is wise to keep apart.

- - - - -

---

## [Author Response · Author response to Decision Letter 1]

3 Jun 2022

Detailed respond is provided in the Rebuttal letter.

---

## [Decision Letter · Decision Letter 2]

17 Aug 2022

PONE-D-22-00156R2

Do aqueous solutions contain net charge?

PLOS ONE

Dear Dr. Ye,

Thank you for submitting your manuscript to PLOS ONE. After careful consideration, we feel that it has merit but does not fully meet PLOS ONE’s publication criteria as it currently stands. Therefore, we invite you to submit a revised version of the manuscript that addresses the points raised during the review process.

Water net charge may be due to molecular polarity of water or a common-ion effect of measurements

for example HCl and NaCl etc... or other known/unknown effect,

as reviewer noted, at this version of manuscript, no any direct data of net charge is presented.

It is better that author change the title to a more appropriate one

for example  Do aqueous solutions contain a semi-net charge? etc...

We look forward to receiving your revised manuscript.

Kind regards,

Abbas Farmany

Academic Editor

PLOS ONE

Journal Requirements:

Additional Editor Comments (if provided):

Reviewers' comments:

Reviewer's Responses to Questions

**Comments to the Author**

1. If the authors have adequately addressed your comments raised in a previous round of review and you feel that this manuscript is now acceptable for publication, you may indicate that here to bypass the “Comments to the Author” section, enter your conflict of interest statement in the “Confidential to Editor” section, and submit your "Accept" recommendation.

Reviewer #3: (No Response)

2. Is the manuscript technically sound, and do the data support the conclusions?

Reviewer #3: Yes

3. Has the statistical analysis been performed appropriately and rigorously? 

Reviewer #3: Yes

4. Have the authors made all data underlying the findings in their manuscript fully available?

Reviewer #3: No

5. Is the manuscript presented in an intelligible fashion and written in standard English?

Reviewer #3: Yes

6. Review Comments to the Author

Reviewer #3: It is an interesting paper with clear experimental setup. However, no any data related to the direct net charge measurement is reported. So, it is better to change the title or prepare the direct net charge data. Also, some figures cited in the text, are missed in the manuscript!

Some part of discussion is better to be removed "Logic implies at least two possible interpretations. The first is that the periodic table may be incomplete, and that some elements may themselves bear charge. The presumption of neutrality

is largely conjectural, and could, in some instances, be inaccurate".

At all, the polarity of water as a polar molecule, creates a slightly negative charge on oxygen and slightly positive charge on hydrogen. This contributes water's properties of attraction. It seems that water charge is generated because oxygen electronegativity instead of hydrogen.

7. PLOS authors have the option to publish the peer review history of their article (what does this mean?). If published, this will include your full peer review and any attached files.

Reviewer #3: No

---

## [Author Response · Author response to Decision Letter 2]

3 Sep 2022

Please see the respond to review file.

---

## [Decision Letter · Decision Letter 3]

27 Sep 2022

Do aqueous solutions contain net charge?

PONE-D-22-00156R3

Dear Dr. Ye,

We’re pleased to inform you that your manuscript has been judged scientifically suitable for publication and will be formally accepted for publication once it meets all outstanding technical requirements.

Kind regards,

Abbas Farmany

Academic Editor

PLOS ONE

Additional Editor Comments (optional):

Reviewers' comments:

Reviewer's Responses to Questions

**Comments to the Author**

1. If the authors have adequately addressed your comments raised in a previous round of review and you feel that this manuscript is now acceptable for publication, you may indicate that here to bypass the “Comments to the Author” section, enter your conflict of interest statement in the “Confidential to Editor” section, and submit your "Accept" recommendation.

Reviewer #3: All comments have been addressed

2. Is the manuscript technically sound, and do the data support the conclusions?

Reviewer #3: Partly

3. Has the statistical analysis been performed appropriately and rigorously? 

Reviewer #3: N/A

4. Have the authors made all data underlying the findings in their manuscript fully available?

Reviewer #3: Yes

5. Is the manuscript presented in an intelligible fashion and written in standard English?

Reviewer #3: Yes

6. Review Comments to the Author

Reviewer #3: This paper addressed a primary hypothesis that aqueous solutions may be charged electrically.

Answering to this question would seem to open a window in our understanding of nature.

The current version of manuscript can be accepted for publication.

7. PLOS authors have the option to publish the peer review history of their article (what does this mean?). If published, this will include your full peer review and any attached files.

Reviewer #3: No

---

## [Editor Report · Acceptance letter]

19 Oct 2022

PONE-D-22-00156R3 

Do aqueous solutions contain net charge? 

Dear Dr. Ye:

I'm pleased to inform you that your manuscript has been deemed suitable for publication in PLOS ONE. Congratulations! Your manuscript is now with our production department. 

Kind regards, 

on behalf of

Dr. Abbas Farmany 

Academic Editor

PLOS ONE